# SVR Chemometrics to Quantify β-Lactoglobulin and α-Lactalbumin in Milk Using MIR

**DOI:** 10.3390/foods13010166

**Published:** 2024-01-03

**Authors:** Habeeb Abolaji Babatunde, Joseph Collins, Rianat Lukman, Rose Saxton, Timothy Andersen, Owen M. McDougal

**Affiliations:** 1Computer Science, Boise State University, Boise, ID 83725, USA; habeebbabatunde@u.boisestate.edu; 2Biomolecular Sciences Graduate Program, Boise State University, Boise, ID 83725, USA; josephcollins177@u.boisestate.edu; 3Department of Chemistry and Biochemistry, Boise State University, Boise, ID 83725, USA; rianatlukman@boisestate.edu (R.L.); rosesaxton@boisestate.edu (R.S.)

**Keywords:** chemometrics, support vector regression, partial least squares, mid-infrared spectroscopy, whey proteins, Kennard-Stones

## Abstract

Protein content variation in milk can impact the quality and consistency of dairy products, necessitating access to in-line real time monitoring. Here, we present a chemometric approach for the qualitative and quantitative monitoring of β-lactoglobulin and α-lactalbumin, using mid-infrared spectroscopy (MIR). In this study, we employed Hotelling T2 and Q-residual for outlier detection, automated preprocessing using nippy, conducted wavenumber selection with genetic algorithms, and evaluated four chemometric models, including partial least squares, support vector regression (SVR), ridge, and logistic regression to accurately predict the concentrations of β-lactoglobulin and α-lactalbumin in milk. For the quantitative analysis of these two whey proteins, SVR performed the best to interpret protein concentration from 197 MIR spectra originating from 42 Cornell University samples of preserved pasteurized modified milk. The R^2^ values obtained for β-lactoglobulin and α-lactalbumin using leave one out cross-validation (LOOCV) are 92.8% and 92.7%, respectively, which is the highest correlation reported to date. Our approach introduced a combination of preprocessing automation, genetic algorithm-based wavenumber selection, and used Optuna to optimize the framework for tuning hyperparameters of the chemometric models, resulting in the best chemometric analysis of MIR data to quantitate β-lactoglobulin and α-lactalbumin to date.

## 1. Introduction

Until the late 20th century, whey was primarily seen as a waste stream derived from cheese production. However, advancements in separation technology, coupled with changing consumer demand for increased protein in foods have led to a surge in demand for whey protein [1]. Today, whey-derived ingredients exhibit the fastest market growth compared to any other dairy ingredient, with a market value of USD 53.8 billion in 2019 and projections to reach USD 81.4 billion by 2025 [2].

Whey protein concentrations present in milk can vary depending on lactation stage, season of milk acquisition, health state of the cow, and cattle breed. The protein content of the colostrum produced initially following birth of a calf contains roughly 70–80% immunoglobulins, which rapidly falls off within days to as low as 1% in the milk. Furthermore, the content of β-lactoglobulin (β-LG) and α-lactalbumin (α-LA) vary widely in colostrum, with ranges of 8 to 30 mg/mL and 8 to 14 mg/mL, respectively [3,4]. Furthermore, the protein concentration continues to vary as lactation proceeds. Ng-Kwai-Hang et al. [5] reported a drop in β-LG concentration from 4.578 to 4.315 mg/mL over the first 60 days of lactation then a steady increase to 4.894 mg/mL at day 365 of lactation. The same study reported a decline in α-LA from 1.773 to 1.441 mg/mL over the same 365-day period. Regester and Smithers [6] noted seasonal variations in β-LG and α-LA present in whey protein concentrates depending on the season of milk collection, and Li et al. [7] reported a drop in α-LA content in milk collected late in the season. Mastitis is also known to alter the concentration of whey proteins in general with a concomitant decrease in both β-LG and α-LA [8]. Milk from different breeds of cattle is also known to show variation in whey protein content. A study from Litwinczuk et al. noted that β-LG varied by ±0.94 mg/mL, and α-LA varied by ±0.13 mg/mL during the summer season in Polish Holstein-Friesian, Jersey, and Simmental cows [9].

The amount of protein in milk, and the concentrations of the individual proteins present, can impact the processing of protein powders, cheese, yogurt, infant milk formula, and more. Whey proteins are prized for their nutritional value as well as their ability to confer functional properties to dairy products such as emulsifying, foaming, viscosity, color and thermal stability, buffering capacity, and gelling [10,11]. Globally, cheese is the most abundant dairy product produced from milk [12]. Traditionally, casein-formed curd is the most common method for cheese making, and the whey fraction, containing the whey proteins that were once discarded, are now integrated back into cheese to improve nutrient value, increase yield, and modify texture [13]. In the production of set type nonfat yogurt, the addition of whey protein into yogurt milk is performed to modify hardness, cohesiveness, and gel elasticity, leading to a more desirable final product [14].

Protein plays a crucial role in the growth and healthy development of human infants [15]. However, the composition of bovine milk is considerably different from human milk in the amount of α-LA present. Bovine milk-based infant formula must be “humanized” by addition of α-LA, because bovine milk only contains about 3.5% α-LA as compared to 22% in human milk [16].

The two most abundant whey proteins in bovine milk are β-LG and α-LA, making up about 50 and 20 percent of the total whey protein composition, respectively. The concentrations of these two proteins individually are of interest to the dairy processing industry. β-LG is a major contributor to the gelling properties of whey because of its high abundance and presence of a free thiol group [17]. Increased β-LG levels have also been noted to cause increased fouling of plate heat exchangers [18]. Furthermore, β-LG is a major allergen of milk, so there is evidence its presence should be limited in certain processing situations. Unlike β-LG, pure α-LA is thermally stable and does not tend to form gels upon heating due to a lack of free thiols to form disulfide bonds [19]. Currently, α-LA is being investigated for use as a carrier of hydrophobic bioactives, like curcumin and capsaicin, in aqueous beverages [20,21,22,23].

The nutritional profile and composition of dairy products are assessed through analysis of protein quantity and quality [24,25]. Traditional protein quantification in the dairy industry has been conducted by the Kjeldahl method, often complemented by high performance liquid chromatograph (HPLC). The Kjeldahl method is a prominent analytical technique for assessing total protein content in dairy products, biological samples, and pharmaceuticals, among others [26,27,28,29,30]. By providing a measure of nitrogen levels in proteins, the Kjeldahl method indirectly quantifies the total protein content. This method involves three fundamental steps: digestion, distillation, and titration [30,31,32]. While Kjeldahl analysis provides accurate measurements of total protein content in milk, it is time-intensive, utilizes harsh chemicals and conditions, and it can only be used to indirectly quantify total protein within a sample, like milk. Conversely, HPLC can be used to quantify individual whey proteins, but the instrumentation is expensive, and specialized technical expertise is required to prepare samples, run the instrument, and assess the results. Rapid quantification of macronutrients in milk (i.e., protein, carbohydrates, and lipids) can be accomplished with infrared spectroscopy (IR), but the data analysis required to deconvolute the resulting spectra and quantify individual proteins has been lacking. Here, we report a chemometric software protocol to quantify the major whey proteins, β-LG and α-LA, in preserved pasteurized modified milk samples, based on the rapid interpretation of mid-infrared spectra (MIR).

There is no rapid, efficient, accurate, and precise method for the quantification of individual whey proteins within milk across the dairy processing industry. Rapid evaluation of milk macro nutritional components, including total protein, total casein, and lipids by IR spectroscopy is, however, commonplace in dairy and other food processing facilities [33,34,35]. Saxton and McDougal [36] explored the application of MIR spectroscopy for qualitative analysis of proteins derived from whey and non-whey sources, employing the amide I/II, lipid, and carbohydrate regions. Their investigation revealed the utility of MIR to detect adulteration of protein powders with inexpensive amino acids, which increases the nitrogen content interpreted by Kjeldahl analysis as falsely correlating to protein quantity in the powders. The MIR detection is dependent on discerning peak shapes within select regions of the spectra, which overlap in complex mixtures like milk, making identification of individual proteins complex. A solution to deconvolute the IR spectra to obtain quantitative and qualitative assessment of individual proteins is the application of chemometrics.

Chemometrics utilize mathematical or statistical methods to select optimal measurement procedures to extract relevant chemical information from chromatographic and spectroscopic data. Chemometrics has emerged as a valuable tool for the interpretation and analysis of complex datasets from gas chromatography, liquid chromatography, and infrared spectroscopy [37,38]. Advances in technology have shown IR-based chemometrics can be used for the rapid and accurate assessment of components in food products including milk, meat, and potato. We detail several relevant studies, across a variety of chemometric techniques, that provided the basis for our investigation.

MIR and chemometrics have been extensively studied for the qualitative and quantitative analysis of pasteurized milk, both for the presence of adulterants, and for the quantification of components such as protein and fat. The key elements of the chemometric work-flow examined in these studies are (1) sample selection approaches, (2) methods for preprocessing the spectral data, (3) wave number selection techniques to improve accuracy and reduce computational complexity, and most often (4) choice of regression algorithm and associated parameter fine-tuning. 

Most studies have focused on the problem of examining the efficacy of various regression techniques for the identification of adulterants in milk products. For example, partial least squares (PLS) and principal component analysis (PCA) were compared for the chemometric analysis of 38 whey protein concentrate (WPC) powders that had been adulterated with milk whey protein (MWP) [39]. The classification of samples into either pure WPC or WPC adulterated with MWP was achieved using PCA, with PLS providing quantification of WPC and MWP, achieving R^2^ values of 99%. 

Mota et al. [40] explored the predictive accuracies of PLS, elastic net (EN), random forest (RF), and gradient boosting machine (GBM) for the quantification of κ-casein from 463 Holstein cows’ milk samples using MIR, finding that GBM outperformed other models in predicting κ-casein with R^2^ value of 81%.

Neto et al. [41] utilized a convolutional neural network (CNN) for binary and multiclass classification analysis of MIR spectra from 4846 milk samples adulterated with sucrose, starch, bicarbonate, peroxide, and formaldehyde. In the same study, GBM and RF were used for similar classification tasks, but with milk constituents including fat, protein, lactose, solids, solids non-fat, casein, milk urea nitrogen, somatic cells counting, freezing point, and sample quality as input variables. The CNN achieved the highest predictive accuracy for both the binary and multiclass classification problems, scoring 99% and 97%, respectively.

Yet another study on adulterants examined PLS, artificial neural networks (ANNs), partial least squares discriminant analysis (PLS-DA), PCA, and support vector machines (SVM), for the identification of milk adulterated with sucrose, urea, and starch [38]. SVM yielded the highest R^2^ values of 98.4%, 97.6%, and 99.6% for starch, urea, and sucrose, respectively. The superior performance of SVM was attributed to superior performance in the handling of both linear and nonlinear relationships within spectral data. 

Dielectric spectroscopy coupled with chemometric techniques including PLS, least-square based SVM (LSSVM), and extreme learning machine (ELM) were used for the quantitative analysis of total protein content in 145 raw fresh milk samples [42]. LSSVM with SNV preprocessing produced the best predictive model and achieved an R^2^ value of 86.5%. In yet another study, PLS, SVR, and ANN chemometric techniques were evaluated for interpretation of MIR to determine the amount of lactoferrin in raw milk [43]. The ANN produced the highest R^2^ value of 60%.

Two spectral preprocessing techniques—Savitzky–Golay (SavGol) with first and second derivatives, and standard normal variate (SNV)—were employed in [40]. SNV preprocessing produced the best predictive model with an R^2^ value of 86.5% for total protein content. In another study, MSC, SNV, weighted multiplicative scatter correction, and inverse multiplicative scatter correction preprocessing techniques achieved the best R^2^ values [42]. Preprocessing techniques including SNV, MSC, SavGol, and mean-centering were evaluated in [43]. The best predictive results were obtained with MSC and mean-centering for the analysis of MIR spectra, and SavGol with a second derivative for NIR spectra.

Some studies perform manual wavenumber selection based on RMSE [41]. Other studies attempt to automate wavenumber selection using various approaches. For example, genetic algorithm-based optimization for wavenumber selection is utilized in [36]. Wavenumber selection is performed by PLS factors in [44].

The majority of studies that report the use of chemometrics in milk IR analysis have not simultaneously and quantitatively analyzed multiple individual whey proteins. Some notable exceptions, include one study that reported the unsuitability of MIR spectroscopy to measure β-LG and α-LA content when employing PLS methodology on raw milk, due to the inability to achieve acceptable prediction accuracy for the two proteins (the best R^2^ values for β-LG and α-LA at 64% and 31%, respectively) [44]. Another study reported better results for the quantification of β-LG and α-LA present at ranges of 0.1–10% with R^2^ value of 99%, but these results were achieved by simplifying their analysis to aqueous whey solutions, rather than raw milk [45]. We hypothesize that a combination of preprocessing and chemometric modeling techniques can be used to overcome the complexity of predicting β-LG and α-LA concentrations from MIR spectra of milk. Here, we report the use of chemometric models to achieve accurate and rapid quantitative analysis of MIR spectra, for the two most abundant whey proteins in milk; β-LG and α-LA. 

## 2. Materials and Methods

### 2.1. Materials, Samples, and Standards

Kaylegion and colleagues [46] generated the first sets of preserved pasteurized modified milk samples in 2006 which we will refer to as Cornell reference samples. They have continued to provide new batches of Cornell reference samples every month since 2006 for use as MIR milk analyzer calibration standards. The Cornell reference sample calibration sets were superior to preserved raw producer milk calibration sets, displaying more consistent inter-day and inter-set calibration slopes than non-modified, raw milk samples [46]. The Cornell reference sets are modified to provide a wider component range and an even distribution of components, as compared to raw milk. These preserved pasteurized modified milk samples have also been used to predict fatty acid chain length and unsaturation level of milk fat by MIR [47], and to calibrate MIR analyzers for the prediction of milk urea nitrogen [48]. Currently, sets of 14 calibration samples are produced on a monthly basis at Cornell University and sent to dairy processors for MIR instrument calibration. Sample sets produced in January, February, and March of 2023 were used to generate a database of 42 unique Cornell reference samples for this study.

The protein standards β-lactoglobulin (≥90%, Catalog #L3908-5G) and α-lactalbumin (≥85%, Catalog #50-176-5110) were purchased from Sigma Aldrich (St. Louis, MO, USA). The amino acid glycine at 99% purity was purchased from Leco.com (Part #502-211) (St. Joseph, MO, USA). The L-lysine monohydrochloride (98.5–100.5%, Catalog #BP386-100) was purchased from Fisher Scientific (Waltham, MA, USA). All chemicals were purchased from Fisher Scientific, including sodium hydroxide pellets (Catalog #S318-500), boric acid powder (Product #A74-1), hydrochloric acid (Catalog #A144S-500), and ammonium sulfate (99.999%, Catalog #AA1063909). Preserved pasteurized modified milk samples (Cornell reference samples) were received from Cornell University (Ithica, NY, USA) on a monthly basis. Fourteen reference samples were in each set that arrived each month for the months of January, February, and March of 2023, for a total of 42 individual samples. Samples were received frozen, packaged on dry ice, and immediately stored at −20 °C until use.

### 2.2. Reagents for the Kjeldahl Method

Unless otherwise stated, all reagents were purchased from Fisher Scientific (Waltham, MA, USA). The reagents used for the Kjeldahl method included concentrated sulfuric acid (95–98%, Product #A484-212), and Kjeldahl catalyst tablets (FisherTab^TM^ CT-37 Kjeldahl Tablets, Product #K3011000); each tablet had a mass of 3.9 g and consisted of 3.5 g K_2_SO_4_ and 0.4 g CuSO_4_. After digestion, 50 mL deionized (DI) water was added to dilute the mixture to prevent precipitation. Solutions (m/v%) of 40% sodium hydroxide, 4% boric acid, 0.1 M sodium hydroxide, and 0.1 M hydrochloric acid were prepared. To 1.0 L of 4% boric acid receiving solution was added 1.5–2.0 mL of a bromocresol green-methyl red mixed indicator (Product #B0120100ML).

### 2.3. Mid-Infrared Spectroscopy (MIR)

Mid-infrared (MIR) spectra were recorded using a Nicolet^TM^ iS20 MIR spectrometer equipped with a Nicolet^TM^ iZ10 module and OMNIC^TM^ 9 software suite (Thermo Fisher Scientific, Waltham, MA, USA). The MIR spectrometer was used in conjunction with an attenuated total reflectance (ATR) diamond plate that was cleaned with isopropanol, allowed to dry, and a background spectrum of nanopure water was recorded prior to sample runs. In each case, the background spectrum was subtracted from the milk sample spectrum to generate a true sample spectrum. Spectrum collection parameters included 1000 scans at a resolution of 2 cm^−1^, with data spacing at 0.482 cm^−1^, using a DTGS KBr detector and KBr beam splitter. Spectra were collected using Blackman–Harris apodization and Mertz phase correction. After data collection, the advanced ATR-correction feature of Thermo Scientific™ OMNIC™ 9 software was applied to all spectra. The Blackman–Harris apodization increases the signal to noise ratio and the Mertz phase correction ensures that a true sample spectrum is generated. The advanced ATR-correction feature makes adjustment for variation in penetration depth and absorption band shift between samples.

### 2.4. High Performance Liquid Chromatography (HPLC)

#### 2.4.1. Sample Handling

Samples were stored at −20 °C until use. For analysis, 1.00 mL of Cornell reference sample was mixed with 200 µL of 10% acetic acid and 200 µL of 1 M sodium acetate, the sample was pH adjusted to 4.3 with HCl. Samples were then centrifuged at 14,000× *g* for 10 min, resulting in three distinct layers. The middle, whey layer was removed and filtered through a 0.45 µm PVDF syringe filter into an amber HPLC vial for analysis.

#### 2.4.2. Chromatography

Chromatography was conducted on an Agilent 1260 Infinity II system with a diode array detector (Agilent Technologies, Santa Clara, CA, USA). A Restek Viva C_18_ column (200 mm × 4.6 mm; 5 um pore size) (Restek, Bellefonte, PA, USA) was used and the diode array detector was set to a wavelength of 214 nm. The mobile phase consisted of two solvents. Solvent A was 0.1% trifluoroacetic acid (TFA) (Sigma Aldrich, St. Louis, MO, USA) in nanopure water and Solvent B was 0.09% TFA in 90% acetonitrile (Fisher Scientific, Waltham, MA, USA) in nanopure water. The gradient began at 42.5% B and increased to 45.0% B at 5 min, then increased to 50% B from 5 to 8 min. From 8 to 9 min solvent B remained at 50%. From 9 to 12 min solvent B increased to 70%. From 12 to 13 min solvent B increased to 100%, and was held at 100% until 14 min. The solvents were returned to starting conditions from 14 min to 16 min. The column was equilibrated at starting condition for an additional 3 min, providing a method with a total runtime of 19 min. 

#### 2.4.3. Calibrations

Standard curves were generated and extraction efficiency was determined using β-lactoglobulin (≥90%, Catalog #L3908-5G) and α-lactalbumin (≥85%, Catalog #50-176-5110) purchased from Sigma Aldrich (St. Louis, MO, USA).

#### 2.4.4. Extraction Efficiency

To determine percent recovery of whey protein extracted from the reference samples, β-LG and α-LA standards were spiked into a native reference sample at concentrations of 0.3 mg/mL and 0.6 mg/mL, respectively. Whey extractions were conducted on both a spiked and unspiked aliquot using the method described and percent recovery of 93% for β-LG and 96% for α-LA, respectively. Extraction efficiency was determined using Equation (1),
(1)$$\%\;\mathrm{Recovery}=\left(\frac{p_{s}}{C_{s}+P_{us}}\right)*100\%$$
where$P_{s}$ is the protein in spiked sample;$P_{us}$ is the protein in unspiked sample;$C_{s}$ is the concentration of spike.

### 2.5. Kjeldahl

The Kjeldahl method was performed using a Foss KT 200 Kjeltec^TM^ (Foss Analytics, Hilleroed, Denmark). The AOAC methods (991.22) and (998.06) were used to determine the protein nitrogen and casein nitrogen content in milk [29,49]. The set of Cornell reference samples was frozen and processed in batches of six. Each milk sample was placed in a water bath and allowed to equilibrate to a temperature of 40 °C. A 5.0–5.1 g portion of each milk sample was immediately pipetted into separate Kjeldahl tubes. For AOAC method 998.06, 70 mL of deionized water and 0.75 mL of acetic acid were added, and a 5-min precipitation period was permitted to separate casein. To ensure full removal of casein, an additional 0.75 mL of sodium acetate was added to each tube followed by filtration through Cytiva Whatman Quantitative Filter Paper: Grade 589/1 circles with a particle retention of 12 to 15 μm. In addition to the milk samples, a blank control tube was also run through the process that did not contain milk, but rather all other reagents. This method is used in quality control to determine the casein content of the milk samples. Casein is of significant interest to the dairy industry because it influences the texture, stability, and nutritional value of dairy products such as cheese and yogurt. AOAC Method 991.22 is used to measure protein nitrogen, which provides accuracy in reporting nutrition content in milk and milk products. The AOAC Method 991.22 procedure begins by addition of 5 mL of deionized water to 40 mL of 15% trichloroacetic acid (TCA), followed by the 5.0–5.1 g portion of each milk sample, left for 5 min, and filtered with a Whatman filter paper. Blank tube contains the filter paper, all Kjeldahl reagents, and no milk. In order to digest the dairy proteins, 25 mL of concentrated sulfuric acid and two Kjeldahl tablets were added to each tube along with the dried filter paper. The tubes were placed in the preheated digestion block at 440 °C for 1 h 45 min. The resulting ammonium sulfate solutions were cooled at room temperature and diluted by addition of 50 mL of deionized water to each tube in preparation for distillation.

In the distillation process, the tube that contains the ammonium sulfate solution and the addition of 50 mL of deionized water is placed in the distillation unit with 100 mL of 4% boric acid in the receiving vessel. Into the ammonium sulfate solution, 100 mL of 40% sodium hydroxide solution (%m/v) was dispensed, followed by 10 min of distillation. During this process, the red receiving solution was observed to change to green. This transformation is due to the conversion of ammonium ions in the refluxing solution into ammonia gas through the distillation process. The ammonia gas is then transferred from the initial solution to the receiving vessel, where it gets captured in an aqueous acidic solution causing the pH dependent color change.

To determine the amount of protein nitrogen that was present in the original sample, the ammonia collected in the receiving solution was titrated with 0.1 M hydrochloric acid. The titration quantifies the amount of ammonia in the receiving solution, leading to a color change from green to light pink within a range of 10–25 mL of 0.1 M hydrochloric acid (indicator pH of 3.70). Equation (2) was used to determine the percentage of nitrogen in each sample. The percent nitrogen value obtained using Equation (2) was multiplied by the conversion factor of 6.38 to give the percent protein value for both true protein (TP) and casein nitrogen (CN) for each sample. By subtracting the casein value from the true protein value, the whey protein content for each sample was determined.
(2)$$\;\%\;Nitrogen=\left(\frac{1.4008*\left(Vs-Vb\right)*M\;}{W\;(g)}\right)$$
where*V_s_* and *V_b_* (mL): Titrant acid used for test portion and blank;*M*: Molarity of the acid solution;*W*(g): Test portion weight. 

### 2.6. Chemometrics Analysis

#### 2.6.1. Data Description

MIR spectra were acquired from three sets of Cornell reference samples, where each calibration set consisted of 14 unique samples. The amount of β-LG and α-LA in each sample was determined using a combination of Kjeldahl and HPLC as described in methods Section 2.4 and Section 2.5. Multiple replicates of each sample were analyzed by MIR spectroscopy to accurately represent milk composition and eliminate instrument fluctuation, leading to a robust MIR calibration of Cornell reference samples [38,45,50]. A total of 212 MIR spectra, representing replicates for each unique sample, were acquired from the original three sets of Cornell reference samples, consisting of 42 unique specimens.

#### 2.6.2. Outlier Detection

MIR spectral consistency within replicates was assessed by the statistical methods Hotelling’s T^2^ combined with Q-residual to identify and omit outliers [51,52,53,54,55]. The outlier detection identified samples that deviate significantly from the majority of spectra using 99% confidence interval of both T^2^ and Q-residual, thereby negatively affecting the predictive ability of the chemometrics model. Hotelling’s T^2^ method is a multivariate statistical technique that simultaneously considers the mean and covariance of the spectra by measuring the variation of each spectrum from the mean of the spectra. Q-residual represents the orthogonal distance of each sample from the prediction of the PLS regression model trained on the remaining spectra. Higher Hotelling T^2^ and Q-residual scores indicate greater deviation from the expected pattern, thus identifying the likelihood that a spectrum is an outlier. By combining the T^2^ and Q-residual measures, outliers that exhibit both extreme values and unusual patterns were identified and omitted from our final dataset. Hotelling T^2^ and Q-residual are mathematically represented in Equations (3) and (4), respectively. From a total of 212 spectra, 15 outliers were identified and subsequently removed from the dataset (Appendix A). 

Hotelling T^2^
(3)$${T_{i}}^{2\;}=\sum_{j=1}^{k}\left(\frac{{t^{2}}_{i,j}}{{s_{j}}^{2}}\right)$$

Q-residual
(4)$$Q={e^{\prime }}_{i}e_{i}$$
where$e_{i}$ is the ith vector in the PLS residual matrix $E=X-TP^{\prime }$;X is the MIR spectra;P is the PLS loadings matrix;T is the PLS scores matrix and $t_{i}$ is its ith vector;k is the number of PLS components used;$s_{j}$ is the standard deviation of jth PLS component.

#### 2.6.3. Data Partitioning

The spectral dataset consisting of 197 spectra was partitioned into a calibration set of 138 spectra (70%) and a validation set of 59 spectra (30%) [38,39]. Two different sample partitioning techniques were employed: the Kennard–Stone algorithm (KS), and random splitting using the scikit-learn library (RS) [36]. To ensure robust model development and evaluation on the limited dataset, leave-one-out cross-validation (LOOCV) was further implemented on the full dataset (calibration and validation sets) [36,37].

KS is a widely used partitioning technique in chemometric analysis, and was used here to generate a calibration set [56,57,58]. The KS algorithm uses the Euclidean distance technique to select samples that span the entire range of the dataset, facilitating the accuracy of chemometric models. The application of KS was tested in three ways: (1) employing the concentration values of α-LA, (2) implement the concentration values of β-LG, and (3) utilizing MIR spectra. Since the concentration values of β-LG and α-LA exhibit a positive linear dependence (see Appendix A), applying either (1), (2), or (3) yields similar results. However, using either (1) or (2) is faster than using (3) because they have fewer data points as compared to (3).

RS was also used to create an alternative calibration set. This technique randomly assigns samples to the calibration set, providing a diverse representation of the data and reduces selection bias that may have been introduced by the KS selection approach [59].

LOOCV is a validation method often applied to small datasets. It was applied here to assess the performance of the KS and RS models. For LOOCV, each sample in the dataset is systematically held out as the validation set, while the remaining samples are used for model training. The leave-one-out process is repeated for each sample in the dataset, ensuring that all samples are used as a test sample. In this study, LOOCV was conducted in two distinct ways: (1) leave-one-replicate-out CV (LOROCV), and (2) leave-one-sample-out CV (LOSOCV). In LOROCV, one replicate of each sample is left out as the validation set while the remaining replicates and samples are used for training. In LOSOCV, all the replicates of each sample are left out as the validation set while the remaining samples are used for training. In the current study, we applied LORO to maximize the use of available data, since we were analyzing 197 spectra. The LORO results may bias the validation set due to the extent of replicate samples in the total, whereas the LOSO approach was expected to perform worse than the LORO due to a lower total number of samples. LORO was viewed as providing an upper bound on performance, while LOSO provides a lower bound. Therefore, the actual performance will likely fall between these two results. The schematic diagram of LOSOCV and LOROCV are presented in Appendix A.

By employing the KS and RS distinct sample partitioning techniques, and LOOCV, we aimed to comprehensively evaluate the performance and generalization capability of the developed chemometric models. The calibration set facilitated model training, while the validation set allowed for unbiased evaluation.

#### 2.6.4. Spectral Preprocessing

Preprocessing was carried out on the calibration (138 spectra) and validation sets (59 spectra) to improve the signal-to-noise ratio of the spectra, and reduce spectra variations that are not relevant for data analysis. To avoid data leakage, the preprocessing techniques were fit on the calibration set and transformed on the validation set. A comprehensive investigation of different preprocessing techniques was conducted to improve the predictive analysis of the spectra data. It was observed in the literature that using multiple preprocessing techniques mostly performs better in making accurate predictions than a single technique, and the order in which these techniques are applied can significantly impact the overall predictive accuracy of the subsequent chemometric analysis [60,61].

The preprocessing techniques that were tested included multiplicative scatter correction (MSC), Savitzky–Golay (SG), mean-centering (MC), normalization, extended multiplicative scatter correction (EMSC), standard normal variate (SNV), robust normal variate (RNV), and local standard normal variate (LSNV). The process was automated using nippy; a preprocessing package for spectral dataset studies [59]. The details of the different preprocessing techniques and the corresponding parameter values explored are summarized in Appendix A.

#### 2.6.5. Wavenumber Selection

The complete MIR spectrum, within the wavenumber range of 4000–400 cm^−1^, is comprised of 14,416 data points. We evaluated wavenumber selection techniques to identify the relevant wavenumbers for the quantification of β-LG and α-LA. The metrics for evaluation included computational time reduction and predictive performance. The techniques assessed were genetic algorithm (GA), interval PLS (iPLS), simulated annealing, PLS coefficient scores, backward interval PLS (BiPLS), and synergy interval PLS (SiPLS). From our survey, the combination of iPLS and GA performed the best to reduce time and yield optimal wavenumber selection results. The initial step employed iPLS according to the protocol of Nørgaard et al. (2000) for (1) the selection of the wavenumbers considered for GA analysis, and (2) the identification of the most relevant interval for the quantification of β-LG and α-LA [62]. This method splits the full spectrum into equidistant intervals and ranks each interval based on its root mean squared error (RMSE) to identify the most important regions for β-LG and α-LA. The GA was then used to make the final wavenumber selections for each protein [63,64]. The GA parameters and iPLS intervals are available in the Appendix A, respectively. The results for simulated annealing, PLS coefficient scores, BiPLS, and SiPLS are given in Appendix A, respectively.

#### 2.6.6. Regression Analysis

A variety of regression techniques including PLS, SVR, ridge, and LR were tested to describe the relationship between the target variables (i.e., concentrations of β-LG and α-LA proteins) and the predictor variables (FT-MIR spectral data). While PLS is widely adopted as an industry-standard method in chemometric analysis due to simplicity, and capacity to assess high dimensional spectra data, SVR is gaining prominence due to aptitude to address both linear and complex non-linear relationships [38]. Ridge and LR are commonly used linear techniques in chemometrics, but like PLS, these methods have limited utility for the analysis of non-linear data.

##### Partial Least Square (PLS)

The partial least squares (PLS) regression method excels at analyzing complex datasets with many variables, by creating a latent space representation of the spectral data and the reference values [43,45]. PLS finds the set of latent variables that retains the most relevant spectral information by capturing the maximum variance between the spectra and the reference values in a lower dimensionality thereby reducing the multicollinearity, redundancy, and dimensionality of the spectral data. PLS is mathematically represented in Equations (5) and (6).
(5)$$X=TP^{\prime }+E$$
(6)$$Y=UQ^{\prime }+F=XB+F$$
whereY is the concentration values of α and β;Q is the PLS scores matrix with respect to Y; F is the residual matrix with respect to Y;B is the PLS regression coefficients.

##### Support Vector Regression (SVR)

The regression technique known as support vector regression (SVR) applies the concepts of support vector machines (SVMs) to regression analysis. SVR operates on a subset of training data points called support vectors, which are essential for creating the regression model. The goal of SVR is to find an optimal hyperplane that maps the input variables (spectral data) to the corresponding continuous output variable (concentration values of β-LG and α-LA), simultaneously maximizing the margin around the training samples and minimizing the prediction error. SVR accomplishes this by providing a tolerance parameter called epsilon, which regulates the margin and provides a limited amount of prediction error tolerance.

The application of kernel functions in SVR enables the identification of complex non-linear relationships between the variables by mapping the input spectra data into higher-dimensional space. SVR can effectively identify linear and non-linear patterns in the data by using several types of kernels, such as linear, polynomial, or radial basis function (RBF). SVR is mathematically represented in Equation (7).
(7)$$min_{w,b{,\xi }_{i}{{,\xi }^{*}}_{i}}\;\;\;\frac{1}{2}||w{\left|\right|}^{2}+C\sum_{i=1}^{N}\left(\xi_{i}+{\xi_{i}}^{*}\right)$$
subject to the constraints
$$y_{i}-w\phi \left(x_{i}\right)-b\;\leq \;\epsilon +\xi_{i}\;\;w\phi \left(x_{i}\right)+b-y_{i}\;\leq \;\epsilon +{\xi^{*}}_{i}\;$$
$\xi_{i},$ ${\xi^{*}}_{i}\geq 0\;\forall \;i=1,...,N$
where$\phi \left(x_{i}\right)$ is the one of linear, polynomial, or RBF kernels;$w\phi \left(x_{i}\right)+b$ is the predicted value;$y_{i}$ is the target output;C is the regularization parameter;$\xi_{i}$ and ${\xi^{*}}_{i}$ are tolerance limits.

##### Ridge Regression

Ridge regression is an extension of linear regression that deals with the problem of multicollinearity through regularization. It reduces the coefficients of less informative wavenumbers toward zero by including a penalty term, alpha in the loss function. The hyperparameter alpha regulates the regularization’s strength; stronger regularization is produced by higher values of alpha. It is mathematically represented in Equation (8).
(8)$$min_{w}\;\;\;||Xw-y|{{|}_{2}}^{2}+\alpha ||w|{|}^{2}$$
where
X is the MIR spectra;w is the ridge regression coefficient vector and $w_{0}$ is the intercept;α is the regularization parameter or penalty term, and α≥ 0. Setting α=0 turn Equation (8) to $min_{w}\;||Xw-y|{{|}_{2}}^{2}$ which is the linear regression cost function;Xw is the predicted concentration value usually denoted by $\hat{y}$;y is the actual concentration value.

## 3. Results

### 3.1. Descriptive Analysis of Protein Content in the Dataset and Spectra Preprocessing

All samples were analyzed by Kjeldahl analysis for true protein, percent casein, and percent whey. The Kjeldahl method is a well-established industry-standard method for quantifying bulk protein in milk, but it cannot quantify individual proteins in the whey fraction. Figure 1 shows a representative chromatogram of a Cornell reference sample with the two target proteins α-LA and β-LG eluting at 8–9 min and 11–12 min, respectively. Based on previous studies, the left shoulder of the β-LG peak is consistent with variant “B” while the right peak is variant “A” [65]. These isoforms differ by two amino acids with sequence differences of D64G and V118A in forms A and B, respectively. As in previous studies quantifying β-LG, areas under the curve for both variants were combined to quantify the total β-LG present [66]. Target proteins eluted with good separation and repeatability with extraction efficiencies of 93% for β-LG and 96% for α-LA, respectively. Extraction efficiency was determined using Equation (1). The RSD of triplicate samples was 0.76 for β-LG and 1.00 for α-LA.

The statistical analysis of protein variability between Cornell reference samples are presented in Table 1. The range of concentrations observed for β-LG and α-LA indicates the diversity of the concentrations of both proteins across the reference set. The ranges noted in our study are consistent with the generally accepted, average values found in bovine milk of 2.0–4.0 mg/mL and 1.5–2.0 mg/mL for β-LG and α-LA, respectively. The ranges of 2.22–4.60 mg/mL and 1.08–2.08 mg/mL for β-LG and α-LA, respectively (Table 1) in our sample data are similar to variations reported in other studies [44].

Some of the raw 197 MIR spectra exhibited significant noise and overlap. To address these challenges, a series of preprocessing techniques were employed, and their effects on the spectra data were systematically evaluated. The preprocessing techniques and the order in which they were applied are presented in Figure 2.

The order in which the preprocessing techniques were applied is as follows: (1) normalization, (2) baseline/scatter correction, and (3) smoothing. The preprocessing workflow in Figure 2 was based on the workflow established in the literature, as explained by Tonolini et al. [45]. Normalization and MC are two common techniques that were considered for scaling the data. Five baseline/scatter correction methods were applied individually to the raw spectra, namely SNV, MSC, RNV, EMSC, and LSNV. The commonly used chemometric preprocessing techniques in milk analysis are MC, SNV, MSC, and SavGol [38,42,45]. EMSC, RNV, and LSNV, which represent modified variations of MSC and SNV, were also introduced. Additionally, instead of the manual approach employed in previous related literature [38,42,44,45], the complex process shown in Figure 2 was automated using nippy to achieve optimal preprocessing. After baseline correction, SavGol smoothing was applied to further reduce noise and enhance spectral resolution. 

The preprocessing results, when simultaneously evaluated for β-LG and α-LA using automated preprocessing, are given in Table 2. PLS regression was employed to evaluate the performance of each combination of preprocessing method to identify the best sequence. The preprocessing combinations were evaluated over the range of n_components, between 1 and 20, to identify the preprocessing techniques that yielded the highest R^2^ score for both β-LG and α-LA. In cases where multiple combinations yielded similar results, the one with the minimum n_components were selected to reduce overfitting on the test samples. From Table 2, the highest $R^{2}$ values ($R^{2}$ = 93%) for β-LG and α-LA were obtained using: (1) MC + normalize + SavGol (filter_window = 151, poly_order = 1, derivative = 0); (2) normalize + SavGol (filter_window = 99, poly_order = 3, derivative = 0); and (3) SavGol (filter_window = 151, poly_order = 2, derivative = 0). The combination of MC, normalize, and SavGol (filter window = 151, polynomial order = 1, derivative = 0) was selected as the optimal preprocessing parameters because it produced the minimum n_components (n_comps = 16) for β-LG and α-LA.

The effects of different preprocessing on the MIR spectra are represented in Figure 3. Figure 3A illustrates the raw spectra as a basis for comparison with the preprocessed data. It was observed that the raw spectra exhibited significant noise especially within then regions 2400–1500 ${cm}^{-1}$. It was observed from Figure 3B,D that the application of SavGol on the raw spectra reduced overlap, resulting in better signal to noise. However, the significance of SavGol was not clearly seen in Figure 3C possibly due to the application of LSNV.

### 3.2. Spectra Interpretation and Regions of Interest

The regions of interest for both β-LG and α-LA identified by GA are presented in Figure 4. Before employing GA, iPLS was initially utilized as an initial step to reduce the number of wavenumbers considered for the GA analysis. Three different options with 20, 25, and 30 equidistant intervals were tested for the iPLS analysis. It was found that the optimal choice was 20 intervals, as it provided superior coverage of the relevant spectral regions, particularly the prominent peaks (amide I, II, and fat) (see Appendix A). The iPLS method results serve as the initial population for the GA. Specifically, the wavenumbers identified by iPLS are included in the initial population of potential features for the GA. This inclusion ensures that the GA begins with a set of candidate features that already exhibit some relevance to the protein concentrations. While the iPLS method provides relevant intervals, it does not provide the specific relevant wavenumber data points within each interval. The GA is used to complement this by further refining the selection to identify the most informative individual wavenumber data points within those intervals.

Based on the results derived from iPLS, wavenumbers within the range of 3000–1000 $cm^{-1}$ were selected as the input for the subsequent GA analysis, and the full spectrum with 14,416 data points was reduced to 10,268 data points.

PLS-based GA was used for the identification of the most relevant wavenumbers for each protein of interest. To achieve this, we binned the iPLS selected data points into 604 bins, each bin representing the summation of 17 contiguous data points (604 × 17 = 10,268). This was performed to reduce the computational time of the GA. Subsequently, the binned data points were subjected to the PLS-based GA. The GA was run for 100 generations, and in each generation, 200 iterations were performed. During the process, the frequency of selection for each wavenumber in each run was recorded. The GA was implemented for β-LG and α-LA separately but with the same GA parameters. The obtained results were then visualized in a bar chart, providing a representation of the wavenumbers’ selection frequency. The selection frequency bar chart is presented in the Appendix A. Out of the 604 binned wavenumbers, 85 bins (i.e., 85 × 17 = 1445 data points) and 51 bins (i.e., 51 × 17 = 867 data points) were selected for β-LG and α-LA, respectively. Figure 4 shows the plot of the spectra with the selected regions for each target protein. The common selected regions for both proteins are wavenumbers within 1800–1700 $cm^{-1}$, 1700–1600$cm^{-1}$, and 1600–1500 $cm^{-1}$. Furthermore, wavenumbers within the regions 1500 $cm^{-1}$ and 3000–2750$\;cm^{-1}$ were selected for β-LG and α-LA, respectively.

### 3.3. Chemometric Models

Four chemometric models, namely PLS, SVR, ridge, and LR were evaluated for their effectiveness in the quantitative analysis of β-LG and α-LA proteins in Cornell reference samples using either (1) the full spectrum without preprocessing (raw spectra), or (2) spectra with the optimal preprocessing obtained and the selected wavenumbers’ data points using GA. The complete parameter spaces for the four models are provided in Table 3.

Since PLS and LR have relatively fewer hyperparameters to optimize, the n_components hyperparameter for PLS and the fit_intercept and copy_X hyperparameters for LR were tuned to achieve the optimal hyperparameters results. However, for SVR and ridge, which have more hyperparameters search spaces, Optuna, an optimization framework for hyperparameter tuning, was utilized to tune their respective hyperparameters [67]. For SVR, the tuned hyperparameters included C, epsilon, kernel, gamma, and degree, while for ridge, alpha, fit_intercept, and solver were optimized. The optimized hyperparameters for linear SVR were found to be (C = 792.3681, epsilon = 0.0311, gamma = 0.0126, degree = 3, and kernel = linear) and (C = 96.3447, epsilon = 0.01069, gamma = 284.4739, degree = 1, and kernel = linear) for β-LG and α-LA, respectively.

The models with the optimized hyperparameters presented in Table 3 were evaluated using root mean squared error for prediction (RMSEP) and coefficient of determination for prediction ($R^{2}P$). The results of each model’s performance are presented in Table 4.

Using raw spectra, the highest $R^{2}P$ values for β-LG and α-LA proteins are 95.3% and 93.0% for KS, 88.8% and 89.7% for RS, 90.7% and 92.1% for LOROCV, and 89.4% and 90.6% for LOSOCV. With optimal preprocessing and GA-selected wavenumbers, $R^{2}P$ values are 96.5% and 94.7% for KS, 89.2% and 90.5% for RS, 92.7% and 92.6% for LOROCV, and 91.9% and 91.8% for LOSOCV. The linear SVR model gave the best results for quantification of both proteins in Cornell reference samples.

## 4. Discussion

The performance of different splitting techniques in chemometrics plays a crucial role in the performance of the predictive models. In our study, we compared the performance of KS and RS using scikit-learn for the quantitative analysis of β-LG and α-LA proteins as presented in Table 4. It was found that KS consistently outperformed RS, providing more accurate predictions, higher $R^{2}$ values (94.7% against 90.5% for β-LG and 96.5% against 89.2% for α-LA), and lower RMSE (0.18 against 0.23 for β-LG and 0.06 against 0.09 for α-LA). This finding aligns with previous research highlighting the effectiveness of KS as a powerful technique for the selection of calibration samples in chemometrics when applied to infrared spectroscopy data [58]. The ability of KS to select representative spectra that capture the variability in the data makes it one of the most preferred calibration sample selection choices for handling high-dimensional spectral data. As a result of this, the use of the KS algorithm for the quantitative analysis of β-LG and α-LA proteins are strongly recommended.

The selection of informative wavenumber regions is a crucial step in analyzing high-dimensional spectral data. It stands to reason that these informative regions would depend on unique structural elements of the proteins of interest. Based on established X-ray crystallography structures of bovine β-LG and α-LA, the two proteins vary significantly in their α-helix and β-sheet compositions. Native β-LG is composed of around 50% β-sheet and 15% α-helix while α-LA is composed of roughly 6% β-sheet and 47% α-helix. The amide I region (1600 to 1690 $cm^{-1}$) and amide II region (1480–1575 $cm^{-1}$) of the MIR spectrum are known to be responsive to protein secondary structures. The amide I region is known to be particularly sensitive to differences in secondary structure with β-sheet components found at 1624–1642 $cm^{-1}$ and α-helix components found at 1656–1663 $cm^{-1}$. The amide II region is less sensitive to secondary structure, but still informative with β-sheets at 1530 $cm^{-1}$ and α-helix at 1545 $cm^{-1}$. In our study, the informative wavenumber regions for predicting the concentrations of β-LG and α-LA and in Cornell reference samples was investigated using GA as presented in Figure 4. Although there is a wider range of wavenumbers in the amide I region, it was found that the wavenumbers in the amide II region were the most informative region for predicting both β-LG and α-LA concentrations, specifically, β-LG at 1520–1560 $cm^{-1}$ and α-LA at 1543–1573 $cm^{-1}$. Furthermore, for both target proteins, the informative regions included wavenumbers within the amide I, and lipid regions. It was also observed that the GA selected wavenumbers in the regions 2500 $cm^{-1}$ and 2750–2900 $cm^{-1}$. Further investigation using iPLS also revealed that the most informative region for both proteins is the amide II region (1480–1575 $cm^{-1}$). This finding is consistent with previous research that highlights the significance of the amide II region for whey protein analysis. There is a strong water band present in milk that may be overlapping with the amide I region and obscuring informative secondary structure information [60,61]. This overlap is not prominent in the amide II region. The identification of these informative wavenumber regions provides valuable insights for analysis of specific milk protein components.

Preprocessing techniques combined with GA search offer potential improvements in predictive modeling. In our study, a combination of preprocessing techniques presented in Table 1 were considered and some of the preprocessed spectra are presented in Figure 3. After evaluating 434 combinations of preprocessing techniques, it was found that the combination of MC, normalization, and zeroth-order SavGol filtering yielded the highest $R^{2}$. By leveraging the GA, the original set of 14,416 spectral data points was narrowed down to a relevant subset for quantifying β-LG and α-LA in Cornell reference sample spectral data. From Table 4, the highest $R^{2}$ values were 95.3% and 93.0% without preprocessing + GA selection, and 96.5% and 94.7% after preprocessing + GA, for α and β, respectively.

We evaluated the performance of linear regression using KS and RS as splitting techniques. Further validation of results was conducted using LOROCV and LOSOCV. LR performed well with KS, achieving satisfactory $R^{2}$ values of 88.7% and 89.5% for β-LG and α-LA, respectively. However, the performance deteriorated when using LOOCV, which might be attributed to the high dimensionality and presence of highly correlated wavenumbers in the milk spectral data. These results emphasize the importance of the need for splitting techniques like LOOCV to ensure reliable model performance especially when working with a small dataset.

The choice of regression models can significantly impact the predictive performance in chemometrics analysis. In our study, we compared the performance of SVR, ridge, LR and PLS regression in modeling the concentrations of β-LG and α-LA (see Table 4). SVR slightly outperformed ridge in terms of $R^{2}$ and RMSEP for both proteins using the KS, RS, and LOOCV. The best $R^{2}$ values achieved using SVR and ridge are (94.7% and 96.5%) and (93.5% and 95.8%) for β-LG and α-LA, respectively. Both SVR and ridge outperformed the other two models: PLS (93.4% and 92.3%) and LR (91.2% and 89.1%). The advantage of SVR in making more accurate predictions highlights its suitability for capturing the complex relationships between the input features and the protein concentration. These findings suggest SVR as a promising regression technique for milk protein analysis. The maximum $R^{2}$ values obtained for β-LG and α-LA using LOOCV are 92.8% and 92.7%, respectively. These results outscored those obtained by Niero et al. [68] who used MIR coupled with uninformative variable elimination and PLS for the analysis of 114 milk samples. The authors employed LOOCV and achieved $R^{2}$ values of 47% and 37% for β-LG and α-LA, respectively. Our study also gave higher predictive results than a study conducted by Bonfatti et al. [44] on the analysis of milk samples using MIR coupled with PLS which reported $R^{2}$ values of 31% and 64% for β-LG and α-LA, respectively.

Our study highlights SVR as the top-performing model for the quantitative prediction of α-LA and β-LG from the interpretation of MIR spectra of milk. Nevertheless, it is important to emphasize that PLS remains a valuable and relevant technique in the realm of chemometrics analysis. This significance stems from PLS’s advantage of having a constrained number of hyperparameters to optimize, which contributes to its practicality and ease of implementation. It is noteworthy that throughout our analysis, PLS served as a complementary and versatile tool to SVR, beyond its role as a predictive model. Specifically, we employed PLS for tasks such as wavenumber selection, the determination of optimal preprocessing techniques, and the identification of outliers. This multifaceted application underscores the utility of PLS in various stages of our analysis, enhancing its value as a fundamental tool in our study.

Recent developments in analytical instruments used to study food include quantum laser cascade (QLC)-based and portable infrared spectrometers that are gaining adoption due to their cost, ease of use, and targeted analysis [69,70,71]. Typically, these instruments have inferior resolution, narrow spectrum range, and lower signal-to-noise than typical benchtop or in-line spectrophotometers [72]. A study by Kappacher et al. [73] compared handheld instruments including the Enterprise sensor from Tellspec, UK, the MicroNIR from Viavi Solutions, and the SCiO from Consumer Physics, with the benchtop NIRFlex N-500 in qualitative analysis of 126 black truffles. Although the benchtop instrument yielded the best results, applying preprocessing techniques to spectra produced from the handheld devices provided results commensurate to the benchtop instrument in most instances. However, these devices did not perform well in some test cases due to their narrow spectral acquisition region, poor resolution, and poor sensitivity. Similarly, a recent study employed a QLC-based MIR instrument for quantitative and qualitative analysis of β-LG, α-LA, and casein in aqueous solutions, spanning the spectra regions of 1470–1730 $cm^{-1}$, covering both amide I and amide II regions [69]. Mean centering and SavGol with first derivative were applied, and PLS was used for calibration. The model achieved RMSECV values of 0.309, 0.302, and 0.426 for β-LG, α-LA, and casein, respectively. While different preprocessing techniques discussed in this paper could be used to improve the spectral quality, there are wavenumbers outside the amide regions identified by the genetic algorithm used in this paper, which are not present in handheld and QLC MIR instruments, thereby limiting the wavenumbers considered for chemometric analysis. This limited wavenumber analysis range is likely to reduce the performance of the developed chemometric techniques. There is a current gap in the literature regarding the application of these emerging instruments to complex mixtures such as milk samples, indicating an exciting area for future exploration and consideration in food analytical studies.

## 5. Conclusions

In conclusion, our study findings demonstrate that MIR coupled with SVR chemometrics proves to be effective for the quantitative analysis of individual proteins in milk. This contrasts with the results reported by Bonfatti et al. and Niero et al., which suggested that MIR coupled with chemometrics cannot accurately quantify individual whey proteins in milk [44,68]. While the previous studies adhered to the well-known industry standard of employing PLS for chemometric analysis of dairy products, we utilized a more complex approach in SVR. Our findings show SVR’s superiority over PLS when assessing β-LG and α-LA protein concentrations in milk, marking a substantial advancement in this domain with $R^{2}$ values of 92.8% and 92.7% for β-LG and α-LA, respectively. Furthermore, we introduced automation into the selection of the optimal preprocessing, distinguishing our methodology from prior studies that utilized manual preprocessing selections. Employing a robust GA-based wavenumber selection technique, we demonstrated its effectiveness in identifying the relevant wavenumbers for β-LG and α-LA protein quantification in milk. The utilization of Optuna, an optimization framework for tuning hyperparameters of chemometric models offers the fast identification of optimal parameters for the chemometric models used in the analysis of β-LG and α-LA proteins.

## Figures and Tables

**Figure 1 foods-13-00166-f001:**
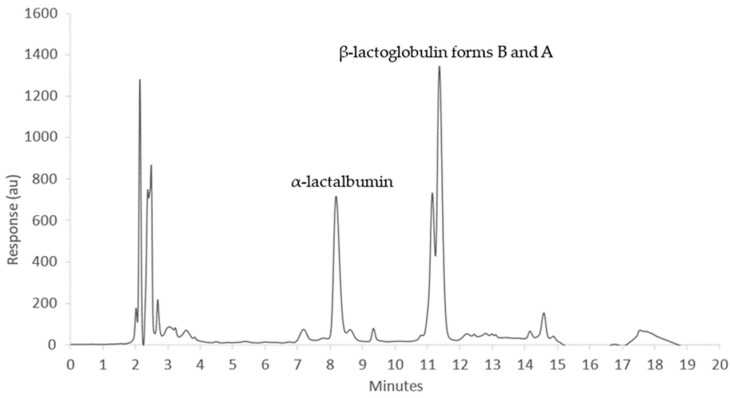
Representative chromatogram of a Cornell reference sample.

**Figure 2 foods-13-00166-f002:**
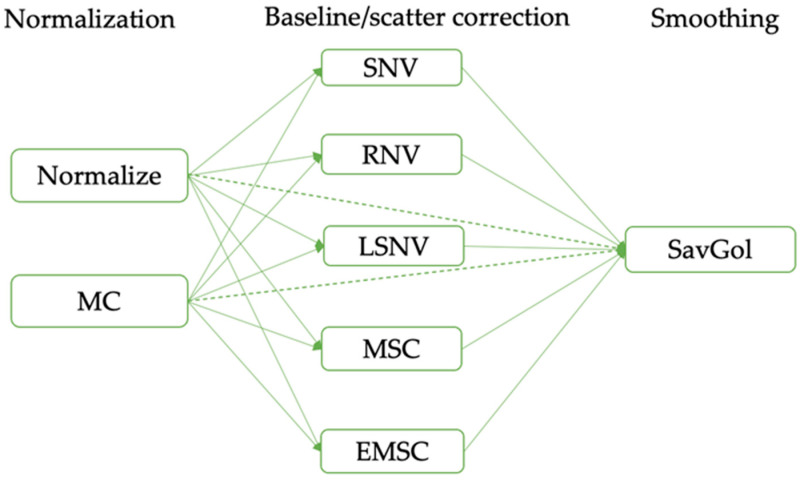
Preprocessing workflow sequences. Dotted green lines represent the combination of normalization and smoothing without baseline/scatter correction, while solid green lines represent the combination of normalization, baseline/scatter correction, and smoothing.

**Figure 3 foods-13-00166-f003:**
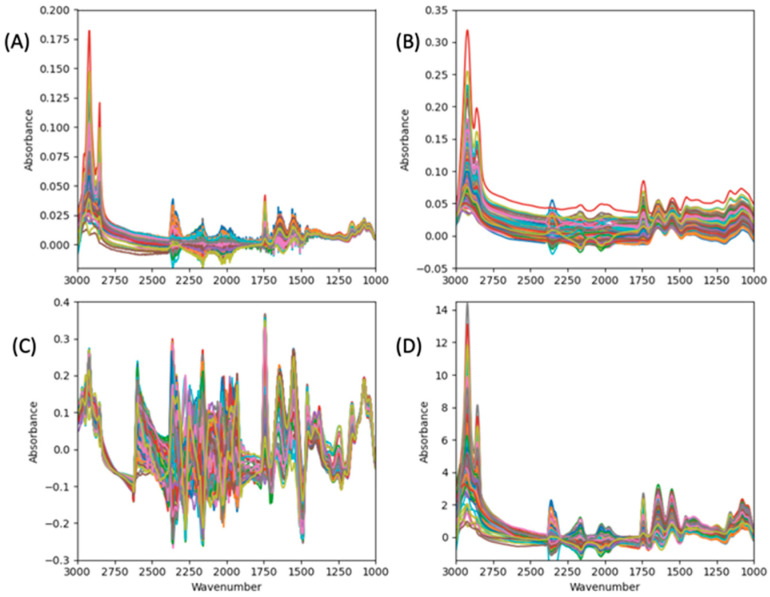
MIR spectra with different preprocessing techniques. (**A**) raw spectra; (**B**) MC + normalize + SavGol (filter_win = 151, poly_order = 1, deriv_order = 0); (**C**) LSNV + normalize + SavGol (filter_win = 151, poly_order = 3, deriv_order = 0); and (**D**) RNV (IQR = 75%, 25%) + SavGol (filter_win = 191, poly_order = 3, deriv_order = 0).

**Figure 4 foods-13-00166-f004:**
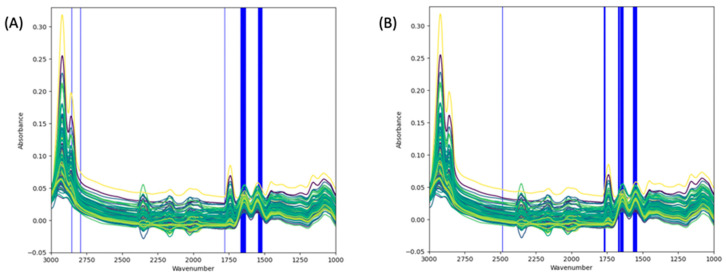
Selected wavenumbers for (**A**) β-LG and (**B**) α-LA using GA. The areas in white background are the excluded wavenumbers and those in blue are the selected wavenumbers. Selected wavenumbers for (1) β-LG: 1520–1560, 1635–1675, 1782–1786, 2796–2800, 2858–2862, and (2) α-LA:1543–1573, 1639–1678, 1760–1778, 2488–2491 cm^−1^.

**Table 1 foods-13-00166-t001:** Quantitative assessment of true protein, casein, and whey percentages determined by Kjeldahl. Individual protein concentrations were determined by HPLC.

Component	Mean	SD	Min.	Max.
True Protein (%)	3.1506	0.6776	2.0034	4.2631
Casein (%)	2.5336	0.5529	1.6120	3.4028
Whey (%)	0.6170	0.1426	0.2763	0.9075
β-LG (mg/mL)	3.3500	0.7600	2.2200	4.6000
α-LA (mg/mL)	1.6000	0.2900	1.0800	2.0800

**Table 2 foods-13-00166-t002:** Evaluation of preprocessing techniques, parameter configuration, predictive accuracies ($R^{2}$) for the quantification of β-LG and α-LA on the validation set using KS splitting, and the selection of the optimal number of PLS components.

Preprocessing	$R^{2}$ β-LG	$R^{2}$ α-LA	n_Comps
Baseline + normalize+			
SavGol(filter_win = 115,	93%	94%	16
poly_order = 1, deriv_order = 0)			
normalize + SavGol(filter_win = 99,	93%	93%	20
poly_order = 3, deriv_order = 0)			
SavGol(filter_win = 115,	93%	93%	20
poly_order = 2, deriv_order = 0)			
LSNV + normalize +			
SavGol(filter_win = 99,	77%	80%	8
poly_order = 0, deriv_order = 2)			
SNV + SavGol(filter_win = 77,	66%	64%	8
poly_order = 3, deriv_order = 0)			
EMSC + SavGol(filter_win = 191,	28%	31%	5
poly_order = 1, deriv_order = 1)			

MC—mean centering; SNV—standard normal variate; MSC—multiplicative scatter correction; SavGol—Savitzky–Golay; EMSC—extended multiplicative scatter correction; LSNV—localize standard normal variate; n_comps—number of PLS components.

**Table 3 foods-13-00166-t003:** Chemometric models and their hyperparameters search spaces tuned by Optuna.

Model	Parameter	Search Space	β-LG_Opt	α-LA_Opt
	C	loguniform(5 × 10^−^³, 1 × 10³)	792.3681	96.3447
	epsilon	uniform (0.01, 0.9)	0.0311	0.01069
SVR	kernel	[‘linear’, ‘rbf’, ‘poly’]	linear	linear
	degree	[1,2,3,4]	3	1
	gamma	Loguniform (1 × 10^−5^, 1 × 10^5^)	0.0126	284.4739
	alpha	Loguniform (1 × 10^−5^, 10)	0.00078	0.00095
		[‘auto’, ‘svd’, ‘cholesky’,		
Ridge	solver	‘lsqr’, ‘sparse_cg’, ‘sag’	lsqr	sparse_cg
		, ‘saga’]		
	fit_intercept	[True, False]	TRUE	TRUE
LR	fit_intercept	[True, False]	TRUE	TRUE
	copy_X	[True, False]	FALSE	FALSE
PLS	n_comps	range (1,20)	14	14

β-LG_opt—optimal parameters selected by Optuna for quantifying β-LG; α-LA_opt—optimal parameters selected by Optuna for quantifying α-LA.

**Table 4 foods-13-00166-t004:** Comparison of models’ performance before and after preprocessing + wavenumber selection using GA. The highest obtained $R^{2}$ values for KS, RS, and LOOCV are in bold.

			KS(P)		RS(P)		LOROCV		LOSOCV	
	**CM**	**Protein**	$R^{2}$	RMSE	$R^{2}$	RMSE	$R^{2}$	RMSE	$R^{2}$	RMSE
Raw	PLS	β−LG	93.00%	0.21	89.70%	0.23	92.10%	0.15	90.60%	0.22
		α−LA	93.80%	0.08	86.80%	0.1	90.70%	0.06	89.40%	0.09
	SVR	β−LG	92.70%	0.21	85.90%	0.28	88.90%	0.18	87.40%	0.26
		α−LA	95.30%	0.07	86.80%	0.1	90.50%	0.06	89.30%	0.09
	Ridge	β−LG	92.40%	0.22	87.50%	0.26	89.50%	0.18	88.00%	0.25
		α−LA	94.20%	0.07	86.80%	0.1	89.80%	0.06	88.60%	0.1
	LR	β−LG	88.70%	0.27	88.90%	0.25	−9.7 × 10^18^	1.3 × 10^8^	−8.7× 10^18^	2.1× 10^8^
		α−LA	89.50%	0.1	88.80%	0.1	−6.7 × 10^18^	6.30 × 10^8^	−1.30 × 10^19^	1.0 × 10^9^
OP+GA	PLS	β−LG	92.30%	0.21	90.00%	0.23	92.60%	0.15	91.70%	0.21
		α−LA	93.40%	0.08	89.00%	0.09	92.20%	0.05	91.10%	0.08
	SVR	β−LG	**94.70%**	**0.18**	**90.50%**	**0.23**	**92.60%**	**0.15**	**91.80**%	**0.21**
		α−LA	**96.50%**	**0.06**	**89.20%**	**0.09**	**92.70%**	**0.05**	**91.90%**	**0.08**
	Ridge	β−LG	93.50%	0.2	90.40%	0.23	92.60%	0.15	91.60%	0.21
		α−LA	95.80%	0.06	88.80%	0.1	92.30%	0.05	91.50%	0.08
	LR	β−LG	81.20%	0.34	85.40%	0.28	89.10%	0.18	88.10%	0.25
		α−LA	90.00%	0.1	86.40%	0.1	91.20%	0.06	90.00%	0.09

P—prediction; CM—chemometric mode; OP + GA—optimal preprocessing + wavenumber selection using GA; KS(P)—KS prediction on validation set; RS(P)—RS prediction on validation set.

## Data Availability

Data is contained within the article or Appendix A.

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
