# Peer review of "SVR Chemometrics to Quantify β-Lactoglobulin and α-Lactalbumin in Milk Using MIR"

_foods, 2024, doi:10.3390/foods13010166_

Round 1

Reviewer 1 Report

Comments and Suggestions for Authors

The paper presents the results of mid-infrared spectroscopy carried out on milk samples. A chemometric data processing makes it possible to quantify two important milk quality: beta-lactoglobulin and alpha-lactal-bumin.

The description of measurements, the results and the data processing are well described and consisted with previsions.

As mid-infrared spectroscopy is an emerging technique for food analytics, thanks to the recent availability of quantum cascade lasers and moderate cost and portable devices, I am kindly asking to the authors to add a paragraph discussing if and to what extent these new instruments could be used for the presented measurements. In particular, if it possible to identify selected wavelengths suitable to achieve if not the same quantitative results, at least qualitative threshold results. This discussion would greatly complete this paper.

Author Response

Reviewer 1 Comments

The paper presents the results of mid-infrared spectroscopy carried out on milk samples. A chemometric data processing makes it possible to quantify two important milk quality: beta-lactoglobulin and alpha-lactalbumin.

The description of measurements, the results and the data processing are well described and consisted with previsions.

Reviewer 1 - Comment 1 (R1 - C1): As mid-infrared spectroscopy is an emerging technique for food analytics, thanks to the recent availability of quantum cascade lasers and moderate cost and portable devices, I am kindly asking to the authors to add a paragraph discussing if and to what extent these new instruments could be used for the presented measurements. In particular, if it possible to identify selected wavelengths suitable to achieve if not the same quantitative results, at least qualitative threshold results. This discussion would greatly complete this paper.

Response to R1 - C1: To address this comment, we have added content to the discussion section. Specifically, we included a new paragraph on page 19, lines 827-851, to describe how our feature selection approach can be transferred to spectra acquired from portable IR devices. Five additional references have been added to support this new information

“Recent developments in analytical instruments used to study food include quantum laser cascade (QLC)-based and portable infrared spectrometers that are gaining adoption due to their cost, ease of use, and targeted analysis [69–71]. Typically, these instruments have inferior resolution, narrow spectrum range, and lower signal-to-noise than typical benchtop or in-line spectrophotometers [72]. A study by Kappacher et al. [73] compared handheld instruments including the Enterprise sensor from Tellspec, UK, the MicroNIR from Viavi Solutions, and the SCiO from Consumer Physics, with the benchtop NIRFlex N-500 in qualitative analysis of 126 black truffles. Although the benchtop instrument yielded the best results, applying preprocessing techniques to spectra produced from the handheld devices provided results commensurate to the benchtop instrument in most instances. However, these devices did not perform well in some test cases due to their narrow spectral acquisition region, poor resolution, and poor sensitivity. Similarly, a recent study employed a QLC-based MIR instrument for quantitative and qualitative analysis of ?-LG, α-LA, and casein in aqueous solutions, spanning the spectra regions of 1470-1730 , covering both amide I and amide II regions [69]. Mean centering and SavGol with first derivative were applied, and PLS was used for calibration. The model achieved RMSECV values of 0.309, 0.302, and 0.426 for ?-LG, α-LA, and casein, respectively. While different preprocessing techniques discussed in this paper could be used to improve the spectral quality, there are wavenumbers outside the amide regions identified by the genetic algorithm used in this paper, which are not present in handheld and QLC MIR instruments, thereby limiting the wavenumbers considered for chemometric analysis. This limited wavenumber analysis range is likely to reduce the performance of the developed chemometric techniques. There is a current gap in literature regarding the application of these emerging instruments to complex mixtures such as milk samples, indicating an exciting area for future exploration and consideration in food analytical studies.”

Reviewer 2 Report

Comments and Suggestions for Authors

-        The authors put a lot of effort into carrying out the scientific work.

-        The authors used milk samples from 2006 as calibration standards for MIR milk analysis and were referred to as Cornell reference sample. The question is whether the contents of these samples did not change during storage, especially β- lactoglobulin and α- lactalbumin in the milk samples.

-        We know that in some countries were used enzyme immunoassays to quantify β- lactoglobulin. The question is why the prepared methods were not compared with enzyme immunoassay.

-        Lines 37 and 38: The authors do not mention all the factors that can affect the protein concentration in milk, such as breed, lactation number and others.

-        The materials and methods section are written clearly and comprehensively.

-        Line 527: Figure 3 was written in the text and number 1 was written below the figure.

-        No figure 4 was mentioned in the results, although figure 3 and 5 are present.

-        There is a problem with the numbering of references in the text:

1-     Line 42: The reference (Regester and Smithers) was written without numbering.

2-     Line 820: The reference (Bonfatti et al., and Niero et al.) was written without numbering.

3-     Line 44: After the name of the reference, the number must be written in the following order, not the date of the reference.

4-     Regarding lines 97, 129, 133 and 182: The reference number must be written directly after the reference name and not at the end of the sentences.

The reference list is fine, only in lines 922, 1025 and 1040 the date of the references need to be written in bold. 

Author Response

Reviewer 2 Comments

Comments and Suggestions for Authors

The authors put a lot of effort into carrying out the scientific work.

Reviewer 2 - Comment 1 (R2 - C1): The authors used milk samples from 2006 as calibration standards for MIR milk analysis and were referred to as Cornell reference sample. The question is whether the contents of these samples did not change during storage, especially β- lactoglobulin and α- lactalbumin in the milk samples.

Response to R2 - C1: We thank the reviewer for identifying this error. The Cornell reference sets that were used for this study are from January, February, and March of 2023. We have corrected the text in the method section, lines 190-193, to read as follows “Kaylegion and colleagues generated the first sets of preserved pasteurized modified milk samples in 2006, that we will refer to as Cornell reference samples [46]. They have continued to provide a new batches of Cornell reference samples, every month since 2006, for use as MIR milk analyzer calibration standards.”, and lines 200-203, to read as follows “Currently, sets of 14 calibration samples are produced on a monthly basis at Cornell University and sent to dairy processors for MIR instrument calibration. Sample sets produced in January, February, and March of 2023 were used to generate a database of 42 unique Cornell reference samples for this study. “

R2 – C2: We know that in some countries were used enzyme immunoassays to quantify β- lactoglobulin. The question is why the prepared methods were not compared with enzyme immunoassay.

Response to R2 – C2: Calibration curves with R2 values of 0.99 or above were generated with HPLC using commercially available standards for β-lactoglobulin and α-lactalbumin. Concentrations of β-lactoglobulin and α-lactalbumin in experimental samples were determined using the HPLC method outlined in the methods section (lines 274-285). An RSD value of 1.00 or less for both proteins across triplicate extractions and quantifications of target proteins from the same experimental sample validated the reproducibility of the method. The HPLC method provided efficient analysis of proteins in samples, and we believe this approach to be better than enzyme-linked immuno assays (ELISA) when more than one protein is being assessed. Using HPLC, each target protein was separated and quantified in a single chromatogram for each sample. The authors are not aware of a commercially available duplex ELISA kit for simultaneous quantification of β-lactoglobulin and α-lactalbumin. While ELISA testing is a common method for quantifying whey proteins in milk, the HPLC method also provides accurate, precise, and reproducible qualitative and quantitative results for whey protein analysis.

R2 – C3: Lines 37 and 38: The authors do not mention all the factors that can affect the protein concentration in milk, such as breed, lactation number and others.

Response to R2 – C3: On pages 1 and 3, lines 37-54, the text has been updated to include variation in lactation stage and breed. Two references have also been added to support the new information. “Whey protein concentrations present in milk can vary depending on lactation stage, season of milk acquisition, health state of the cow, and cattle breed. The protein content of the colostrum produced initially following birth of a calf contains roughly 70-80% immunoglobulins, which rapidly falls off within days to as low as 1% in the milk. Furthermore, the content of ?-lactoglobulin (?-LG) and α-lactalbumin (α-LA) vary widely in colostrum, with ranges of 8 to 30 mg/mL and 8 to 14 mg/mL, respectively [3,4]. Furthermore, the protein concentration continues to vary as lactation proceeds. Ng-Kwai-Hang et al., reported a drop in ?-LG concentration from 4.578 to 4.315mg/mL over the first 60 days of lactation then a steady increase to 4.894 mg/mL at day 365 of lactation [5]. The same study reported a decline in α-LA from 1.773 to 1.441 mg/mL over the same 365-day period. Regester and Smithers noted seasonal variations in ?-LG and α-LA present in whey protein concentrates depending on the season of milk collection, and Li et al., reported a drop in α-LA content in milk collected late in the season [6,7]. Mastitis is also known to alter the concentration of whey proteins in general with a concomitant decrease in both ?-LG and α-LA [8]. Milk from different breeds of cattle is also known to show variation in whey protein content. A study from Litwinczuk et al., noted that ?-LG varied by ± 0.94 mg/mL, and α-LA varied by ± 0.13 mg/mL during the summer season in Polish Holstein-Friesian, Jersey, and Simmental cows [9]. “

R2 – C4: The materials and methods section are written clearly and comprehensively.

Response to R2 – C4: We thank the reviewer for their feedback on this section.

R2 – C5: Line 527: Figure 3 was written in the text and number 1 was written below the figure.

Response to R2 – C5: The figure number has been changed in the text from "Figure 3" to "Figure 1" on line 538.

R2 – C6: No figure 4 was mentioned in the results, although figure 3 and 5 are present.

Response to R2 – C6: The figure numbers has been changed. "Figure 5" is now labeled "Figure 4" on lines 627 and 644.

R2 – C7 a - d: There is a problem with the numbering of references in the text:

  1. Line 42: The reference (Regester and Smithers) was written without numbering.

Response to R2 – C7a: The reference for Regester and Smithers has been updated in the text as citation [6] on line 49 of page 2.

  1. Line 820: The reference (Bonfatti et al., and Niero et al.) was written without numbering.

Response to R2 – C7b: Reference numbers have been updated as follows: Bonfatti et al., is [44], and Niero et al., is [68] on line 860.

  1. Line 44: After the name of the reference, the number must be written in the following order, not the date of the reference.

Response to R2 – C7c: The date has been removed from reference to Li et al., and the citation number has been written after the author names, see line 48.

  1. Regarding lines 97, 129, 133 and 182: The reference number must be written directly after the reference name and not at the end of the sentences.

Response to R2 – C7d:  References have been updated as follows:

Line 97: updated from Saxton and McDougal to Saxton and McDougal [36] on line 105

Line 129: updated from Mota et al. to Mota et al. [40] on line 137

Line 133: updated from Neto et al. to Neto et al. [41] on line 141

Line 182: updated from Kaylegion and colleagues to Kaylegion and colleagues [46] on line 190

R2 – C8: The reference list is fine, only in lines 922, 1025 and 1040 the date of the references need to be written in bold. 

Response to R2 – C8: All references have been updated with bold reference dates as recommended by the reviewer.